# Accelerated MCMC for Satellite-Based Measurements of Atmospheric CO$_2$

**Otto Lamminpää [1],\*** , **Jonathan Hobbs [2]** , **Jenný Brynjarsdóttir [3]** , **Marko Laine [1]** , **Amy Braverman [2]** , **Hannakaisa Lindqvist [1]** and **Johanna Tamminen [1]**

[1]   Finnish Meteorological Institute, 00560 Helsinki, Finland
[2]   Jet Propulsion Laboratory, California Institute of Technology, Pasadena, CA 91125, USA
[3]   Department of Mathematics, Applied Mathematics and Statistics, Case Western Reserve University, Cleveland, OH 44106-7058, USA
\*   Correspondence: otto.lamminpaa@fmi.fi

**Abstract:** Markov Chain Monte Carlo (MCMC) is a powerful and promising tool for assessing the uncertainties in the Orbiting Carbon Observatory 2 (OCO-2) satellite's carbon dioxide measurements. Previous research in comparing MCMC and Optimal Estimation (OE) for the OCO-2 retrieval has highlighted the issues of slow convergence of MCMC, and furthermore OE and MCMC not necessarily agreeing with the simulated ground truth. In this work, we exploit the inherent low information content of the OCO-2 measurement and use the Likelihood-Informed Subspace (LIS) dimension reduction to significantly speed up the convergence of MCMC. We demonstrate the strength of this analysis method by assessing the non-Gaussian shape of the retrieval's posterior distribution, and the effect of operational OCO-2 prior covariance's aerosol parameters on the retrieval. We further show that in our test cases we can use this analysis to improve the retrieval to retrieve the simulated true state significantly more accurately and to characterize the non-Gaussian form of the posterior distribution of the retrieval problem.

**Keywords:** OCO-2; Markov Chain Monte Carlo; carbon dioxide; aerosols; retrieval; uncertainty quantification

## 1. Introduction

One of the greatest challenges to the future of planet Earth is climate change arising from global warming, which in turn is accelerated by human-originated emissions of greenhouse gases. The major warming effects are due to atmospheric carbon dioxide (CO$_2$), which is emitted in increasing amounts since industrialization. A majority of these emissions comes from e.g., usage of fossil fuels in transport, in manufacturing and in agricultural industries. In order to model climate change reliably and to predict future scenarios, accurate carbon flux estimates are crucial. In addition to source estimation, this also means quantifying the natural sinks of carbon, e.g., uptake by oceans and forests. Fluxes are estimated by a procedure called inverse modeling, which in addition to computational models needs observations of atmospheric greenhouse gas concentrations. While notable ground based measurement networks exist for this purpose (e.g., GLOBALVIEW-CO2: Cooperative Atmospheric Data Integration Project—Carbon Dioxide for in-situ, flask, and aircraft measurements [1] used in flux inversions, and TCCON [2,3] for total atmospheric column measurements), their coverage is not extensive enough to account for all of the global phenomena relating to the carbon cycle and in particular CO$_2$ emissions.

One way to improve the coverage and resolution of the measurements is to globally collect air column-averaged measurements of CO$_2$, denoted X$_{CO2}$, using satellites. Measurements from orbit offer a

great coverage around the globe, but special care is needed with quantifying the accuracy and precision of these measurements. Importantly, systematic errors that vary in space and/or time can lead to errors in flux estimates, which are inferred from $CO_2$ gradients. Simulation experiments with inversion systems (e.g., [4,5]) indicate that regional or seasonal biases of a few tenths of a part per million (ppm) can induce substantial errors in flux estimates. Although local $CO_2$ variations in the lower troposphere close to sinks and sources can be large, these variations are confined to low altitudes and thus have a relatively low contribution to the total variability of $X_{CO2}$. NASA's Orbiting Carbon Observatory 2 (OCO-2) satellite mission ([6–9]) aims to satisfy these requirements and has been operationally measuring $X_{CO2}$ since July 2014.

From orbit, OCO-2 measures radiances of reflected solar radiation, that is, absorption of sunlight in Earth's atmosphere on different wavelengths. These measurements are inverted to get an atmospheric state $x$ that describes conditions on the light path, such as $CO_2$ concentrations and aerosol and surface properties. The resulting inverse problem is solved using a retrieval algorithm, which consists of a state-of-the-art full physics model that extensively describes the related physical phenomena in the atmosphere, and the so called Optimal Estimation (OE) framework [10], in which the algorithm minimizes the cost function related to the posterior distribution of the retrieval. This is obtained via statistical formulation of inverse problems, where in short error bars are thought of as probability distributions, and the famous Bayes' Formula is used to regularize the problem by implementing prior knowledge of the underlying statistics and physics (operational implementation: ACOS Algorithm [7,9]). OE is is an efficient tool for computing maximum A posteriori (MAP) estimates and related uncertainties, but these quantities rely on a Gaussian approximation of the posterior distribution. It is, however, clear that the forward model in this case is non-linear, which is known to result in a non-Gaussian posterior.

Non-Gaussian posterior distributions can be explored using Markov chain Monte Carlo (MCMC) methods, in which a set of samples is created such that they are distributed along the desired posterior distribution. Quantities of interest, e.g., mean, variance, distribution shapes etc. can then further be inferred from the samples. MCMC was first introduced in atmospheric retrievals by [11], the advantages of MCMC method for validating operational satellite retrievals were demonstrated using real Envisat/GOMOS observations in [12]. Other work includes [13–15]. An application of MCMC to the OCO-2 retrieval problem was recently performed by [16] as a simulation study following the approach of [17] with a simplified and computationally efficient surrogate forward model that captures all the essential effects of the Full Physics Model's $CO_2$, surface and atmospheric aerosol parameters. It was noticed that results from MCMC and OE can differ from each other, and furthermore in some cases not even agree with the simulated ground truth.

It was further discussed in [9,18] that perturbations in the first guesses of aerosol part of the state vector lead to unpredictable changes in retrieved $X_{CO2}$. The retrievals in the study were performed using simulated OCO-2 measurements ([19]) with a known ground truth using the full physics model and OE. As a linear problem would have a unique minimum for the cost function, this suggests that the aerosol parametrization in the model causes problems in the retrieval that might significantly affect the accuracy of retrieved $X_{CO_2}$ values. This is due to scattering by aerosols being a substantial source of non-linearity in the forward model (as was found for example by [17,20]), which might translate to a non-Gaussian posterior distribution.

In addition to the potential effect of aerosol parametrization of the forward model, it was found in [16] that the MCMC on OCO-2 surrogate model has a very low acceptance ratio (number of accepted samples over number of computed samples), and that chains can often converge extremely slowly. These are symptoms of non-linearity, high dimensionality and strong correlations between the parameters in MCMC. Recent work on dimension reduction for MCMC in Bayesian inverse problems (e.g., [21,22]) that are solved using MCMC sampling has successfully addressed this problem. This approach is called the Likelihood

Informed Subspace (LIS) dimension reduction and it is based on identifying a subspace of the state space that contains all the information that is possible to obtain from the measurement. It was shown in [15] that this methodology actually aligns with the informative directions formalism by [10] that is widely used as a standard in the atmospheric remote sensing community.

In this work we focus on the analysis of single retrievals and their posterior distributions. We implement the LIS dimension reduction for the OCO-2 surrogate model retrieval to speed up the convergence of MCMC algorithm and further to characterize the non-Gaussian parts of the full posterior distribution of the state vector, particularly in the parameters relating to the aerosols in the physics model. In this study, we use the same simulated radiances and OE retrievals as in [16]. The rest of the article is organized as follows: in Section 2, we present an overview of the OCO-2 measurement and surrogate forward model, as well as introducing the MCMC and dimension reduction framework used in this study. Section 3 summarizes our results comparing the dimension-reduced MCMC with a full dimensional one, results related to the form of the posterior distribution, and results of an experiment with a relaxed retrieval prior covariance. Finally, Section 4 contains discussion of the results, conclusions and further research topics.

## 2. Methodology

In this section, we give an overview of NASA's Orbiting Carbon Observatory 2 satellite and the simplified surrogate forward model used to investigate uncertainties of the actual operational retrieval algorithm. We then describe briefly the Markov Chain Monte Carlo method and the Adaptive Metropolis-Hastings implementation used in this study. Lastly, we introduce the mathematical basis for Bayesian formulation of inverse problems as well as the principles of using the low information content of the OCO-2 measurement to construct a Likelihood-Informed Subspace dimension reduction scheme for accelerating our MCMC simulations.

### 2.1. Remote Sensing of $Co_2$ by OCO-2

We will give a brief overview of the instrument since this is crucial in understanding the nature of the inverse problem at hand. The OCO mission is described in detail by [6,8,23].

The OCO-2 instrument is composed of three spectrometers that measure light reflected from Earth's surface in the near-infrared part of the spectrum at three separate wavelength bands. These bands are centered around 0.765, 1.61 and 2.06 μm and are called the $O_2$ A-band, the weak $CO_2$ band and the strong $CO_2$ band, respectively. Each observation furthermore consists of 1016 radiances from each band, and together they form the observed sounding, denoted as $y \in \mathbb{R}^{3048}$ in this work. The satellite flies in a polar, sun-synchronous orbit that covers the whole Earth with a 233 orbit 16-day repeat cycle, during which it collects measurements with a surface footprint of less than 2.25 km down-track and 1.3 km cross-track every 0.333 s.

### 2.2. Surrogate forward Model

We consider the inverse problem of estimating an unknown atmospheric state vector $x \in \mathbb{R}^n$ from observed radiances $y \in \mathbb{R}^m$,

$$y = F(x) + \varepsilon, \tag{1}$$

where the *forward model* $F : \mathbb{R}^n \rightarrow \mathbb{R}^m$ describes the physics relating the atmospheric state to measured radiances and the random variable $\varepsilon \in \mathbb{R}^m$ is the observation error caused by forward model approximations and instrument noise.

The state-of-the-art Full Physics Model describes the light absorption and scattering by trace gases and aerosols in the atmosphere. While this is a comprehensive way to compute atmospheric radiative transfer,

the model is computationally heavy to evaluate and hence is not suitable for Monte Carlo experiments. We ease this problem by following the approach in [16] and use the surrogate forward model by [17] to account for the major sources of error in the retrieval. A more comprehensive look can be found in Appendix A.2, but in short, the state vector used in the surrogate model can be split into four parts:

- **$CO_2$ vertical profile:** The model considers the amount of molecules in 20 fixed pressure levels of the atmosphere. The concentration of $CO_2$ as well as pressure and temperature are taken to be constant inside each level.

- **Surface pressure:** A single parameter of the state vector is retrieved to help identify the total amount of molecules in an atmospheric column. Since $O_2$ has a near constant concentration in each atmospheric layer, the total absorption on the $O_2$ band can be used to estimate the total amount of air at the measurement location.

- **Surface albedo:** Earth's surface reflects radiation differently at different wavelengths, and the relation of reflected vs. total incoming radiation is given by albedo. In the surrogate model this corresponds to 6 parameters: a Lambertian albedo and it's spectral slope at each of the 3 wavelength bands of the observed absorption spectrum.

- **Aerosol parameters:** Small particles in the atmosphere absorb and scatter light in complex ways, which is taken into account by the model as an aerosol model. The vertical profile of aerosol concentration is modeled with 3 parameters per aerosol type, first of which is the logarithm of Aerosol Optical Depth (AOD) at 755 nm which describes the total intensity of aerosol effects. The second parameter describes the height of the maximum aerosol concentration, and the third describes the thickness of the aerosol layer, with a small value corresponding to a thin layer. The aerosol profile calculated from these parameters is proportional to

$$AOD \exp\left(-\frac{(x - x_a)^2}{2\sigma^2}\right),$$

(2)

where $x_a$ is the layer height and $\sigma$ is the layer thickness. In this work the number of atmospheric aerosol types is 4, which amounts to a total of 12 aerosol parameters in the state vector.

The main quantity of interest in the OCO-2 inverse problem is the column-averaged dry air mole fraction of $CO_2$, denoted $X_{CO_2}$, which is a weighted average $CO_2$ concentration from the first 20 parameters of the state vector. Motivated by this, we partition the state vector as

$$x = \begin{bmatrix} x_\alpha \\ x_\beta \end{bmatrix},$$

(3)

where $x_\alpha$ contains the $CO_2$ profile and $x_\beta$ has the rest. We can now calculate $X_{CO_2}$ as

$$X_{CO_2} = h^T x_\alpha,$$

(4)

where $h \in \mathbb{R}^{20}$ is a pressure weighting function [23] assigning appropriate weights to the $CO_2$ profile.

As noted in [17], the surrogate model does not include several elements of the operational retrieval's state vector, including a temperature profile offset, a water vapor profile scale factor, band-specific dispersion parameters for wavelength offsets, band-specific empirical orthogonal function scale factors, and solar-induced fluorescence (SIF) in the $O_2$A-band. As such, the purpose of the surrogate model is to act as a computationally feasible means for quantifying the propagation of uncertainty in input arguments into the algorithm outputs. Based on these insights, further more quantitative experiments can then be carried out for the Full Physics model.

### 2.3. Bayesian Formulation of the Inverse Problem

We follow the Bayesian approach [24] to solving inverse problems, which gives us a solution and the related uncertainties as a statistical distribution of states $x$ conditioned on observed data $y$ and regularized by a choice of prior distribution. This is obtained through Bayes' formula as

$$\mathbb{P}(x|y) \propto \mathbb{P}(y|x)\mathbb{P}_{pr}(x),\qquad(5)$$

where $\mathbb{P}(x|y)$ is the posterior distribution, $\mathbb{P}(y|x)$ the likelihood and $\mathbb{P}_{pr}(x)$ the prior distribution. The proportionality $\propto$ comes from a constant that does not depend on the unknown $x$. In this work, we assume the prior to be Gaussian, which is denoted as $\mathcal{N}(x_0, S_{pr})$, i.e.

$$\mathbb{P}_{pr}(x) \propto \exp\left(-\frac{1}{2}(x-x_0)^T S_{pr}^{-1}(x-x_0)\right).\qquad(6)$$

For comparability, we use the same prior mean $x_0$ and covariance $S_{pr}$ as the operational OCO-2 retrieval. Also, the additive noise is assumed to be zero-mean Gaussian with known covariance matrix, $\varepsilon \sim \mathcal{N}(0, S_{obs})$. This way, the likelihood function will have the form

$$\mathbb{P}(y|x) \propto \exp\left(-\frac{1}{2}(y-F(x))^T S_{obs}^{-1}(y-F(x))\right).\qquad(7)$$

In this work, we use the same error statistics as [17], where the variance of $y$ for the diagonal $S_{obs}$ is given by

$$\mathrm{Var}(y_{i,j}) = c_j F_{i,j}(z),\qquad(8)$$

where $z \in \mathbb{R}^n$ is a simulated state vector, $j = 1, 2, 3$ is the index of wavelength band, $i = 1, \ldots, m_j$ is the index of wavelengths in given band $j$ and $c_j$ denotes a band specific constant to yield broadly comparable signal-to-noise ratios to the operational OCO-2 retrieval. This parameterization is a simplification relative to the operational retrieval, which includes channel-specific parameters based on instrument calibration [25].

To solve this inverse problem, also referred to as *retrieval*, a common strategy in atmospheric remote sensing is to use an optimization algorithm such as Levenberg-Marquadt (see e.g., [10]) solver to find the posterior mode. This corresponds to minimizing the cost function

$$-2\log\mathbb{P}(x|y) = (y-F(x))^T S_{obs}^{-1}(y-F(x)) + (x-x_0)^T S_{pr}^{-1}(x-x_0).\qquad(9)$$

This procedure is often called *Optimal Estimation* (OE). We denote the minimizer obtained with OE by $\hat{x}$. On top of this, OE yields a linear Gaussian approximation of the posterior covariance, denoted $\hat{S}$, such that

$$\hat{S} = \left(K(\hat{x})^T S_{obs}^{-1} K(\hat{x}) + S_{pr}^{-1}\right)^{-1},\qquad(10)$$

where $K_{ij} = \frac{\partial}{\partial x_j} F_i$ is the Jacobian of the forward model.

### 2.4. Markov Chain Monte Carlo

While OE can provide a fast and computationally inexpensive solution to the retrieval problem, it is still a linear Gaussian approximation of the true posterior. In the case of OCO-2 retrieval, the forward model is non-linear and hence we need a way to characterize the actual underlying distribution. When the forward model is non-linear, the posterior distribution can be explored by Markov chain Monte Carlo (MCMC) sampling. In this work, we follow [16] and utilize adaptive Metropolis-Hastings MCMC

from [26]. The algorithm progresses by moving from state $x_t$ to a proposed state $x_{t+1}$ drawn from a proposal distribution $\mathcal{N}(x_t, C_t)$ with a density denoted $q(x, \cdot)$. The new point is then accepted or rejected with probability

$$\min\left(1, \frac{\mathbb{P}(x_{t+1}|y)q(x_{t+1}, x_t)}{\mathbb{P}(x_t|y)q(x_t, x_{t+1})}\right) \tag{11}$$

In the adaptive Metropolis algorithm, we are additionally updating the proposal covariance matrix by empirically calculating the covariance from the already obtained chain:

$$C_t = \begin{cases} C_0, & t < t_0, \\ s_d\text{cov}(x_o, \ldots, x_{t-1}) + \epsilon\mathbb{I}, & t \geq t_0 \end{cases} \tag{12}$$

where $t_0$ is the training length and $s_d$ is a scaling parameter from [26] that optimizes the sampling efficiency. Also, as was noted in that article, we extend the adaptation length to updating the covariance every 100th time step after the initial $t_0$.

## 2.5. Likelihood-Informed Subspace

In atmospheric remote sensing the information content of the measurement is an important concept to be considered e.g., when designing the instruments and constructing the retrieval methods. Most of the time, the actual amount of informative directions in the measurement is small compared to the dimension of the state vector. This is discussed in detail by Rodgers [10]. Consistent with this idea, the OCO-2 Algorithm Theoretical Basis Document (ATBD) [23] states that the $CO_2$ part of the state vector only has 2 degrees of freedom for the signal. Motivated by these observations, we use the likelihood- informed subspace (LIS) Dimension reduction ([21,22]) to truncate the dimension of the inverse problem in order to speed up the MCMC computation. It was shown in [15] that initial derivation of LIS is equal to Rodgers' discussion on informative directions of the measurement, so we present the basics of the formulation using the Rodgers' formalism.

We identify a subspace of the state space that has all the measurable information of the retrieval. This can be found by first linearizing the inverse problem and then using linear transformations to rotate and scale the space so that we can see how much of the natural variability of $x$, coming from the prior, can actually be distinguished from the noise $\varepsilon$. This is accomplished by first scaling the problem so that we end up with unit covariances for prior and error, and then diagonalizing the resulting whitened problem. We can then see how many independent degrees of freedom we can obtain from our measurement for the state $x$ by finding the directions of variance greater than unity, since the rest are indistinguishable from the measurement noise.

Consider a linearized version of the inverse problem in Equation (1),

$$y = K(x - x_0) + \varepsilon, \tag{13}$$

where $K$ denotes the Jacobian matrix of the forward model with elements $K_{ij} = \frac{\partial}{\partial x_j}F_i$. Next, we use the Cholesky factorizations of prior and error covariances,

$$S_{pr} = \mathcal{L}_{pr}\mathcal{L}_{pr}^T, \quad S_{obs} = \mathcal{L}_{obs}\mathcal{L}_{obs}^T, \tag{14}$$

and perform pre-whitening of the problem by setting

$$\widetilde{y} = \mathcal{L}_{obs}^{-1}y, \quad \widetilde{K} = \mathcal{L}_{obs}^{-1}K\mathcal{L}_{pr}, \quad \widetilde{x} = \mathcal{L}_{pr}^{-1}(x - x_0) \text{ and } \widetilde{\varepsilon} = \mathcal{L}_{obs}^{-1}\varepsilon. \tag{15}$$

Pre-whitening transforms the error and prior distributions to have zero mean and unit covariance, that is, $\widetilde{\varepsilon} \sim \mathcal{N}(0, \mathbb{I})$ and $\widetilde{x} \sim \mathcal{N}(0, \mathbb{I})$. Now the problem can be written as

$$\widetilde{y} = \widetilde{K}\widetilde{x} + \widetilde{\varepsilon}. \tag{16}$$

We now consider the prior predictive distribution for whitened measurements $\widetilde{y}$, obtained by evaluating the linearized forward model at states $\widetilde{x}$ drawn from the whitened prior distribution with added white noise. It follows that $\widetilde{y}$ will be distributed with covariance

$$\widetilde{S}_y = \mathbb{E}[\widetilde{y}\widetilde{y}^T] = \mathbb{E}[(\widetilde{K}\widetilde{x} + \widetilde{\varepsilon})(\widetilde{K}\widetilde{x} + \widetilde{\varepsilon})^T] = \widetilde{K}\widetilde{K}^T + \mathbb{I}. \tag{17}$$

To get rid of possible non-diagonal elements in the resulting covariance matrix, we further diagonalize the scaled problem by using the singular value decomposition of the whitened Jacobian,

$$\widetilde{K} = W\Lambda U^T. \tag{18}$$

We denote this scaling by a superscript $'$. This gives us the diagonalized problem

$$\begin{aligned} y' = W^T\widetilde{y} = &\ W^T\widetilde{K}\widetilde{x} + W^T\widetilde{\varepsilon} \\ = &\ \Lambda U^T\widetilde{x} + W^T\widetilde{\varepsilon} \\ = &\ \Lambda\widetilde{x}' + \widetilde{\varepsilon}'. \end{aligned} \tag{19}$$

The transformations $\varepsilon'$ and $x'$ conserve the unit covariance matrices for the scaled error and prior distributions, since $W^TW$ and $U^TU$ both result in unit matrix. It follows that the scaled data $y'$ is distributed with covariance $\Lambda^2 + \mathbb{I}$. This is a diagonal matrix with unit contribution from the noise in each diagonal element. From this covariance we can deduce that only the variability coming from states $\widetilde{x}$ corresponding to singular values $\lambda_i$ of scaled Jacobian $\widetilde{K}$ that are greater than unity can be distinguished from the measured data; the rest is masked by measurement noise. In other words, the number of singular values greater than unity correspond to the number of informative directions or degrees of freedom for signal in the measurement. Furthermore, degrees of freedom for signal and noise are invariant under linear transformations [10], and as the unknown $x$ and the error $\varepsilon$ are assumed to be independent, the same result is also valid for the original $y$.

On the other hand, the linear approximation of the posterior distribution given by Equation (16) yields the following expression for the posterior covariance of $\widetilde{x}$:

$$\widetilde{S}_{post} = \left(\widetilde{K}^T\widetilde{K} + \mathbb{I}\right)^{-1}. \tag{20}$$

Using the same reasoning as before, we see that only the directions in the likelihood corresponding to singular values of scaled Jacobian that are greater than unity can be obtained from the measurement, and the rest are given by the prior. This leads to a connection with the work in [21,22], given also by the fact that $\widetilde{K}\widetilde{K}^T$ has the same eigenvalues as $\widetilde{K}^T\widetilde{K}$, all of which are the squared singular values of $\widetilde{K}$. The matrix $\widetilde{K}^T\widetilde{K}$ in turn is the same thing as the prior-preconditioned Gauss-Newton Hessian

$$\widetilde{H} := \mathcal{L}_{pr}^T H \mathcal{L}_{pr} = \mathcal{L}_{pr}^T K^T S_{obs}^{-1} K \mathcal{L}_{pr} = \widetilde{K}^T\widetilde{K} \tag{21}$$

from [21]. Here, $H$ denotes the Gauss-Newton approximation of the Hessian of $-2\log\mathbb{P}(x|y)$.

Informative directions of the measurement can now be identified with the first $r$ eigenvectors of $\widetilde{H}$ corresponding to the first $r$ eigenvalues greater than unity. These are the first $r$ columns of matrix $V$ given by

$$\widetilde{H} = V\Lambda^2 V^T. \tag{22}$$

We now use the informative directions to reduce the dimension of the inverse problem. Consider the low-rank approximations for the posterior of the form

$$\widetilde{\mathbb{P}}(x|y) \propto \mathbb{P}(y|P_r x)\mathbb{P}_{pr}(x), \tag{23}$$

where $P_r$ is rank $r$ projection matrix. In [21,22] it was shown that the optimal $P_r$ that minimizes the Hellinger distance between rank $r$ approximations and the full posterior is given by the eigendecomposition of $\widetilde{H}$ as follows:

**Definition 1.** *[LIS]: Let $V_r \in \mathbb{R}^{n \times r}$ be a matrix containing the first r eigenvectors of the prior preconditioned Gauss-Newton Hessian $\widetilde{H}$. Define*

$$\Phi_r := \mathcal{L}_{pr}V_r \text{ and } \Theta_r := \mathcal{L}_{pr}^{-T}V_r. \tag{24}$$

The rank $r$ *LIS projection* for the low-rank posterior approximation in Equation (23) is given by

$$P_r = \Phi_r\Theta_r^T. \tag{25}$$

The range $\mathbb{X}_r$ of projection $P_r : \mathbb{R}^n \to \mathbb{X}_r$ is a subspace of state space $\mathbb{R}^n$ spanned by the column vectors of matrix $\Phi_r$. We call the subspace $\mathbb{X}_r$ the *likelihood-informed subspace (LIS)* for the linear inverse problem, and its complement $\mathbb{R}^n \setminus \mathbb{X}_r$ the *complement subspace (CS)*.

**Definition 2.** *The matrix of singular vectors $V = [V_r V_\perp]$, where $V_\perp \in \mathbb{R}^{n \times n-r}$ contains the rest of the eigenvectors not included in $V_r$, forms a complete orthonormal system in $\mathbb{R}^n$ and we can define*

$$\Phi_\perp := \mathcal{L}_{pr}V_\perp \text{ and } \Theta_\perp := \mathcal{L}_{pr}^{-T}V_\perp \tag{26}$$

*and the projection $\mathbb{I} - P_r$ can be written as*

$$\mathbb{I} - P_r = \Phi_\perp\Theta_\perp^T. \tag{27}$$

*Define the LIS-parameter $x_r \in \mathbb{R}^r$ and the CS-parameter $x_\perp \in \mathbb{R}^{n-r}$ as*

$$x_r := \Theta_r^T x, \quad x_\perp := \Theta_\perp^T x. \tag{28}$$

*The parameter x can now be naturally decomposed as*

$$\begin{aligned} x &= P_r x + (\mathbb{I} - P_r)x \\ &= \Phi_r x_r + \Phi_\perp x_\perp. \end{aligned} \tag{29}$$

*Using the previous definitions, we can write the approximate posterior as*

$$\widetilde{\mathbb{P}}(x|y) = \mathbb{P}(y|\Phi_r x_r)\mathbb{P}_{pr}(x_r)\mathbb{P}_{pr}(x_\perp). \tag{30}$$

*2.6. MCMC Sampler*

When dealing with a non-linear forward model, the Hessian matrix depends on the point $x$ and thus is not constant on the state space. It was suggested in [21] that a way to get around this issue is to take a Monte Carlo average of $\widetilde{H}$ over some reference distribution. The effect of different distributions was further explored in [27]. Since we have obtained the posterior mode and the related covariance approximation from the OE retrieval, we use these to get an average

$$\hat{H}_N = \mathcal{L}_{pr}^T \left( \frac{1}{N} \sum_{k=1}^{N} H(x^{(k)}) \right) \mathcal{L}_{pr}, \tag{31}$$

where the $N$ samples $x^{(k)}$ are drawn from $\mathcal{N}(\hat{x}, \hat{S})$. We then use the eigenvalue decomposition $\hat{H}_N = \hat{V}\hat{\Lambda}^2\hat{V}^T$ to find a basis for the global LIS analogously to the linear case. We then proceed to sample $x_r$ from the reduced posterior $\mathbb{P}(y|\Phi_r x_r)\mathbb{P}_{pr}(x_r)$, and add a sample drawn from the Gaussian complement prior $\mathbb{P}_{pr}(x_\perp)$ to get an approximation of the full posterior.

In adaptive MCMC, a good starting value for $C_0$ as a training covariance is $s_d\hat{S}$. However, since we already have computed the expected value of the Gauss-Newton Hessian in the LIS computation, we can use this Monte Carlo estimator to better capture the variability over the non-linear posterior and set

$$C_0 = s_d \left( \bar{H} + S_{pr}^{-1} \right)^{-1}, \tag{32}$$

where $\bar{H}$ is the expectation of the Gauss-Newton Hessian over the reference distribution $\mathcal{N}(\hat{x}, \hat{S})$. Next, we project $C_0$ into LIS and use that as a starting covariance for the MCMC. The projection is given by

$$\Theta_r C_0 \Theta_r^T := \widetilde{C}_r \in \mathbb{R}^{r \times r}. \tag{33}$$

After dimension reduction, the MCMC sampling then proceeds with the standard Adaptive Metropolis algorithm where we draw at time step $t$ a new candidate point from proposal $\mathcal{N}(x_{t-1}, C_t)$. Acceptance probability is computed by first projecting the LIS parameter $x_r$ into full space as

$$\Phi_r x_r + x_0 \in \mathbb{R}^n. \tag{34}$$

The likelihood function is then evaluated using the full-dimensional vector as an input to the surrogate model. Finally, after completing the MCMC chain, we draw a sample with the same length as the chain from the complement prior

$$x_\perp \sim \mathcal{N}(0, \mathbb{I}), x_\perp \in \mathbb{R}^{n-r}. \tag{35}$$

The sample is then projected back to full space as

$$\Phi_\perp x_\perp \in \mathbb{R}^n \tag{36}$$

and added to the LIS-chain. It follows from (23) that this way we get a sample from the optimal rank $r$ posterior approximation, as can be seen from the $CO_2$ profiles in Figure 3.

## 3. Results

We present our results in 3 subsections. The example soundings we use are chosen from the data used in [16], which is based on the work in [17]. In short, this study used a clustering method to group the first 18 months of global OCO-2 measurements into 100 geolocation templates based on physical, climatological

and temporal properties. Within a template, synthetic data was created by randomly sampling a Gaussian mixture model, the parameters of which were empirically gained from OCO-2 $CO_2$ profiles and MERRA climatology. We specifically chose soundings from a template in South America (centered at Southern Bolivia) in which the initial results of OE and MCMC in [16] did not agree. The dominating aerosol species for this template are sulphate and dust (see the dominating aerosol type map in [23]). First, we focus on comparing the LIS implementation and its results with previous MCMC results from [16], as well as showing the relative contributions of LIS and CS to the sampling, as well as improvements achieved in computational speed of the MCMC when using LIS. In the second section we inspect the shapes of posterior marginal distributions of model aerosol parameters and compare the MCMC results with OE. Lastly, we perform MCMC retrievals for three example cases using both the operational OCO-2 prior covariance matrix and a modification of it, in which the prior has been widened to allow a greater variability on aerosol layer width parameters.

### 3.1. LIS Implementation

For constructing the LIS basis for the retrieval problem, we use the posterior approximation obtained from OE and sample 200 points from $\mathcal{N}(\hat{x}, \hat{S})$ which are then used to compute the Monte Carlo average (see Section 2.6). As seen in Figure 1, the total number of eigenvalues greater than unity is found to be 18, which is in accordance with [23] stating that the degrees of freedom for signal in the full physics model is around 20. Moreover, the number of eigenvalues greater than unity can be seen to converge to 18 even when using 100 samples, so using 200 samples will reasonably ensure this convergence. In order to ensure that all the variability is captured by LIS we set $r = 19$ as the effective dimension of the problem, as the 19th eigenvalue is still relatively close to 1. Our tests showed that using less than 19 eigenvectors resulted in a significantly different posterior distribution than the full dimensional case, whereas using 19 or more resulted in the same posterior as in the full dimensional case.

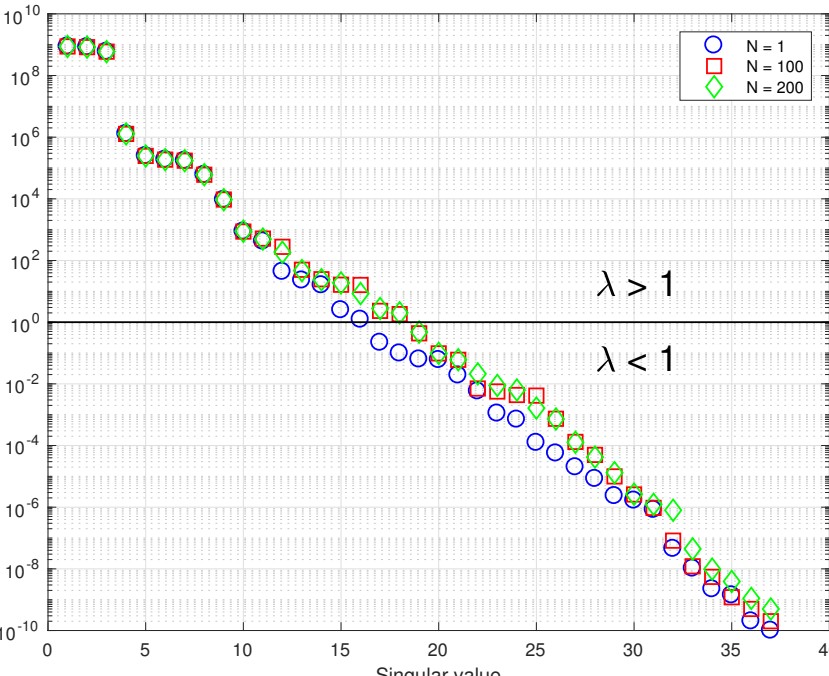

**Figure 1.** Eigenvalues of the Hessian matrix with different numbers of samples used in the calculation of the Monte Carlo average 2.6. The eigenvalues are separated to less than and greater than unity by a black line.

As an example case, we start the comparative MCMC chain from a value that significantly disagrees with the simulated true value of $X_{CO_2}$. We perform the full dimensional MCMC with a chain length of 250,000 samples with a training time of 1000 samples as was done in previous work. For comparability, we perform the dimension reduced MCMC runs using the same sample sizes. The resulting $X_{CO_2}$ chains for both runs are shown in Figure 2. We also show the histograms of posterior $X_{CO_2}$ for both runs as well as the simulated ground truth value. When comparing the last 100 000 samples of the chains, we get acceptance rates of 1.7% for the full-dimensional MCMC and 5.1% for LIS MCMC, and effective sample sizes (number of samples that can be considered uncorrelated; see e.g., [28], pp. 5–7) of 54 and 134, respectively. The scaling parameter for adaptive MCMC was set to $s_d = 0.3 \times 2.4^2/d$, where $d$ is the dimension of the estimated parameter. This choice led to a better acceptance rate for the chain which is better for data visualization, but did not otherwise affect the efficiency of the sampler.

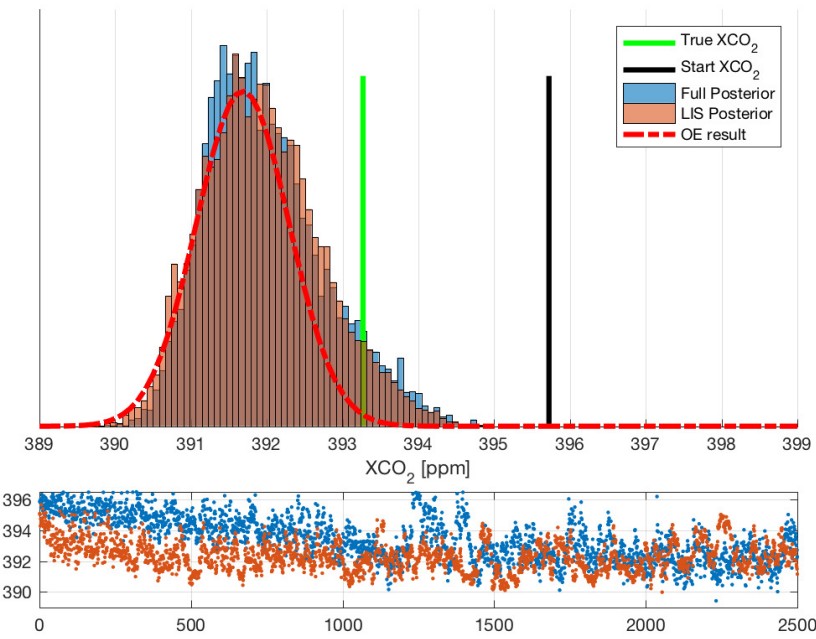

**Figure 2.** Upper panel: $X_{CO_2}$ posterior histogram from full dimensional MCMC (blue) compared to LIS MCMC (red). Also shown the starting value for MCMC and simulated ground truth (green). Lower panel: every 100th sample of $X_{CO_2}$ chain from full dimensional MCMC (blue) and LIS MCMC (red).

Although the OCO-2 algorithm retrieves a full CO2 profile on 20 levels, it only validates its full-column retrievals, as there are typically only two degrees of freedom for $CO_2$ information in the retrieved profile. Nevertheless, we show the Optimal Estimation's retrieved 20 pressure levels in the left panel of Figure 3 plotted against the MCMC results and the ground truth. The contributions from the LIS parameter $x_r$ and CS parameter $x_\perp$ projected back to the full space are also shown in the right panel of Figure 3. Together, these results show that the LIS implementation works well and succeeds in capturing all the information present in the measured data with an increased computational efficiency. Furthermore, dimension reduced MCMC seems to give a decent estimation of the $CO_2$ profile as well.

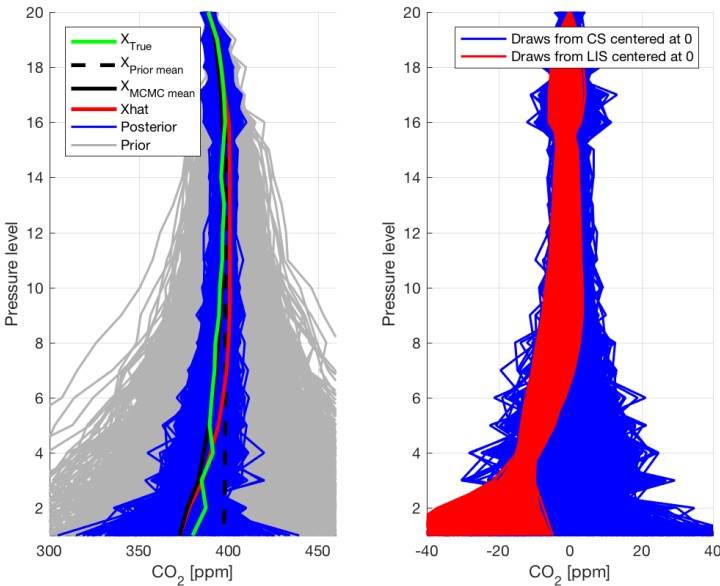

**Figure 3.** Left panel: $CO_2$ profiles from the MCMC retrieval compared with prior mean, ground truth and OE retrieval result. Right panel: $CO_2$ parts of samples from LIS and CS projected back to full space.

### 3.2. MCMC Results

The previous subsection offers an opportunity to conduct further analysis with dimension reduction accelerated MCMC. With the MCMC approach we can compute the full posterior distribution, and a natural question is whether the OE optimization algorithm actually finds the mode of the posterior distribution and how well this corresponds to the expected value obtained from MCMC sampling.

In the studied case the initial termination criteria for OE suggested that the cost function did not have a unique minimum, but our tests concluded that by utilizing the information of the MCMC run, the OE algorithm could be tuned so that differing starting values did actually converge to same minimum given enough computation time. Details of this parameter tuning can be found in Appendix A.1. The parameter tuning we performed achieved an approximate agreement with MCMC mean and $\hat{x}$ estimate obtained from OE, but this resulted in significantly more iterations required for the optimization. However, this effect is not further addressed in this work as we focus on comparing the results of a well-performing optimization algorithm and MCMC.

Next, we perform MCMC simulation for the same example sounding as before (from the South American template from [16]) using the improved optimization algorithm parameter values from Appendix A.1 to initialize the MCMC with a more reliable OE. Figure 4 shows the marginal posterior distributions for all state vector aerosol parameters. A clear non-Gaussian shape is detected in 1D marginal posterior distributions of dust and cloud liquid water AOD parameters, which also correlates to the OE not being able to pick up the full posterior width. The true value of the aerosol parameters is relatively poorly captured by both MCMC and OE mean estimates. Especially the parameters for sulphate and water coefficient 2 (aerosol altitude) have the true value outside the edge of the posterior histogram, which means that these parameters were not identified by MCMC.

We further illustrate the non-Gaussian behavior by plotting the 2D distributions of these state vector elements in Figure 5, and the uncertainties coming to $X_{CO_2}$ from aerosol parameters as 2D distributions in Figure 6. Both of these figures show that the posterior estimate as 95% confidence intervals obtained from OE is consistently not able to contain the full uncertainty in these parameters as shown by the MCMC

simulation. Compared to OE, the probability region given by MCMC still contains the true value in most cases, which means that the uncertainty estimate from OE is again inadequate. We can further see that parameters 29 and 38 (sulphate and water altitudes) have the MCMC posterior tail away from the true value, which further shows that these directions are not identified and cause error to $X_{CO_2}$.

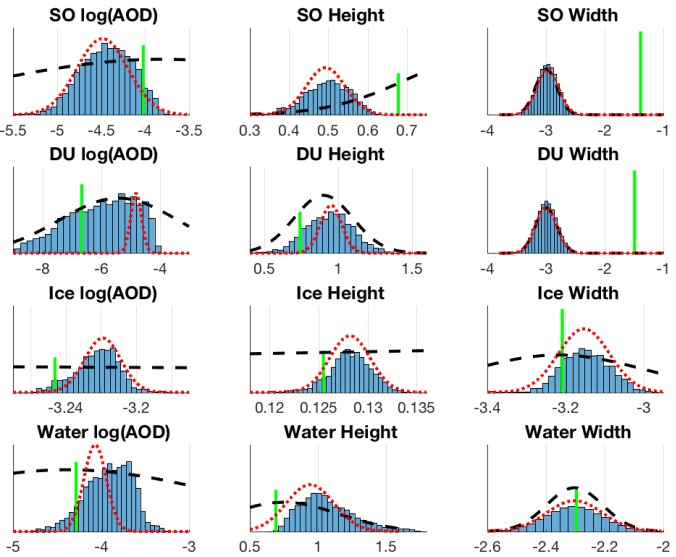

**Figure 4.** Aerosol parameter marginal histograms with operational prior (black dashed line), ground truth (green) and OE retrieval result (red dotted line). SO = sulphate, DU = dust, Ice = cloud ice and Water = cloud liquid water.

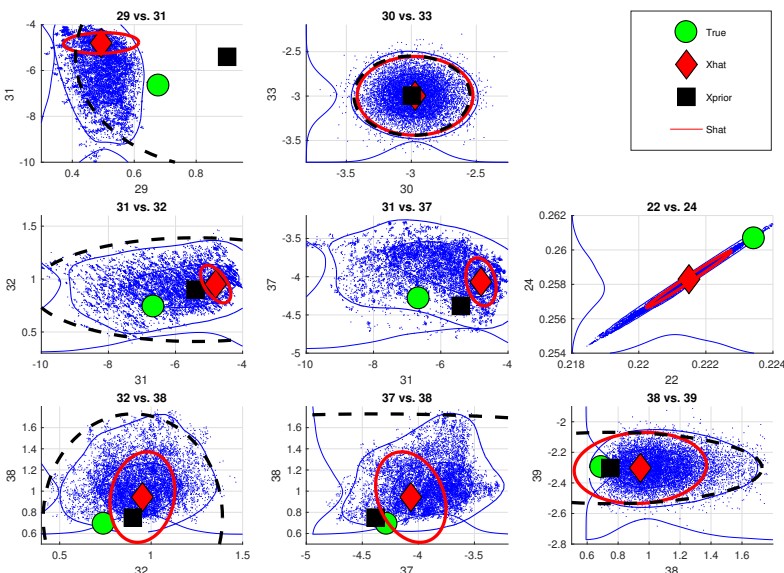

**Figure 5.** 2D posterior distributions of selected aerosol and surface parameters from LIS MCMC (blue with 95% posterior confidence interval on blue contour) compared with ground truth (green), prior (black with 95% confidence interval on black ellipse) and OE (red with 95% posterior confidence interval on red ellipse). Parameter numbers agree with the numbering in the surrogate model description (see Table A3).

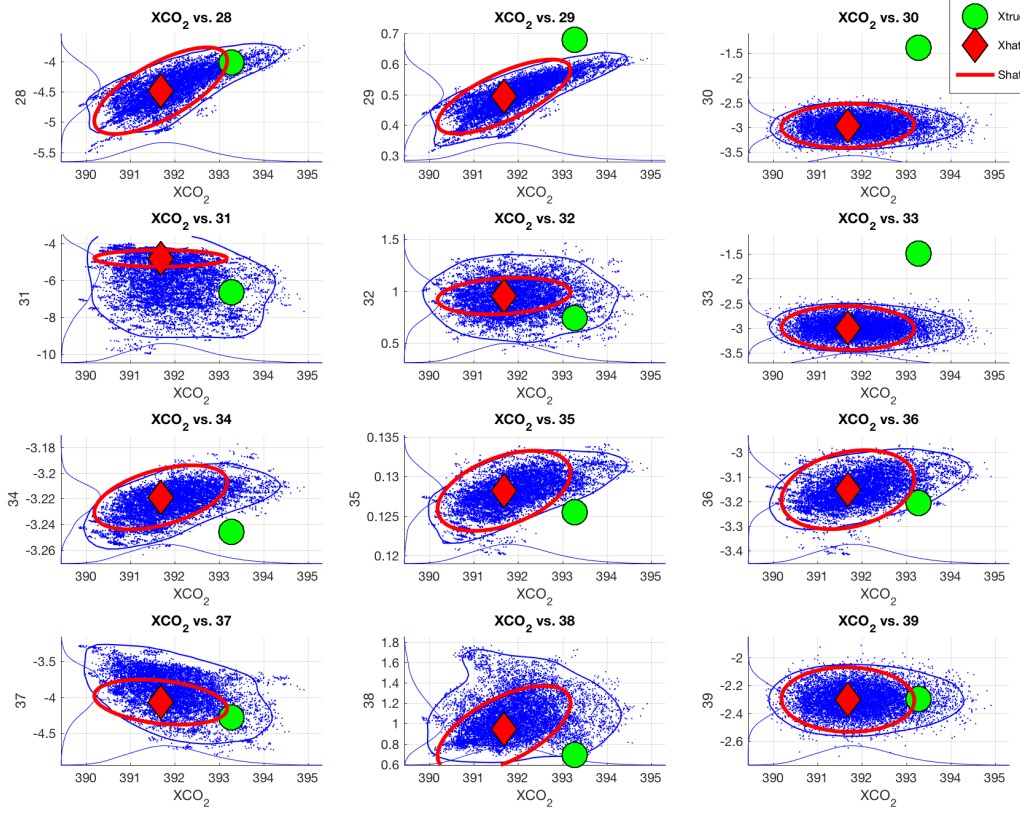

**Figure 6.** 2D posterior distributions of $X_{CO_2}$ against aerosol parameters from LIS MCMC (blue with 95% posterior confidence interval on blue contour) compared with ground truth (green) and OE (red with 95% posterior confidence interval on red ellipse). Parameter numbers agree with the numbering in the surrogate model description and correspond to those in Figure 4.

### 3.3. Wider Prior

In addition to gaining information about the shape of the posterior distribution, we can see from Figure 5 that the simulated state vector's parameters 30 and 33 (corresponding to sulphate and dust layer thickness parameters) are outside the prior distribution induced by the operational prior. Since the state vectors were simulated using an empirical covariance derived from a credible ensemble of simulated and meteorological data ([17], Section 4.1), this phenomenon might possibly occur in the operational OCO-2 retrieval as well, because both retrievals use the same prior. It is worth noting that in the operational retrieval, the prior constraint on the aerosol width is so tight that this parameter is effectively not retrieved. This was based on early work [29] which showed that the spectra had little sensitivity to it.

Focusing on the 3 parameters describing the sulphate aerosol cloud, the posterior correlation matrix of $\hat{S}$ in Figure 7 shows that parameters 28 and 29 (sulphate log AOD and aerosol altitude) have a strong correlation with the $CO_2$ part of the state vector. Additionally, we can deduce that if the aerosol profile width (parameter 30) is outside the prior range and hence cannot be retrieved, the retrieval might try to compensate for this with the remaining sulphate parameters. This in turn could cause the observed MCMC mean in Figure 2 to not align with the simulated ground truth. To test this hypothesis, we first select 2 additional example soundings from the same template as before such that they as well have state

vector parameter 30 valued outside the operational prior distribution. We then perform a second round of OE and MCMC but this time we increase the entry in the prior covariance's diagonal element 30 so that the prior is in effect non-informative for this parameter, allowing variability that captures the true simulated value. Furthermore, since the prior is diagonal for the aerosol parameters, any cross-correlation adjustment is not necessary.

We illustrate the observed effect of widening the prior covariance to the retrieved $X_{CO_2}$ in Figure 8. The $X_{CO_2}$ computed from both chains (using operational (blue) and widened (red) prior) are plotted in comparison with the corresponding OE results and the simulated ground truth. We can clearly see that the wider prior covariance had, in these cases, the effect of significantly improving the accuracy of both MCMC and OE retrievals of $X_{CO_2}$. In Figures 9–11, we have summarized the MCMC histograms, for each test case, of parameters relating to log AOD, height, and width of the modelled aerosol layer for all 4 aerosol types present in the retrieval. We also show the ground truth, prior distribution and OE results for comparison. The parameter SO Coefficient 3 on the upper right panel is the one with widened prior, and in comparison to previous histograms in Figure 4, we see that this parameter is now retrieved relatively well. As we can see, this change also had the effect of improving the overlap of resulting posterior histograms and the true value.

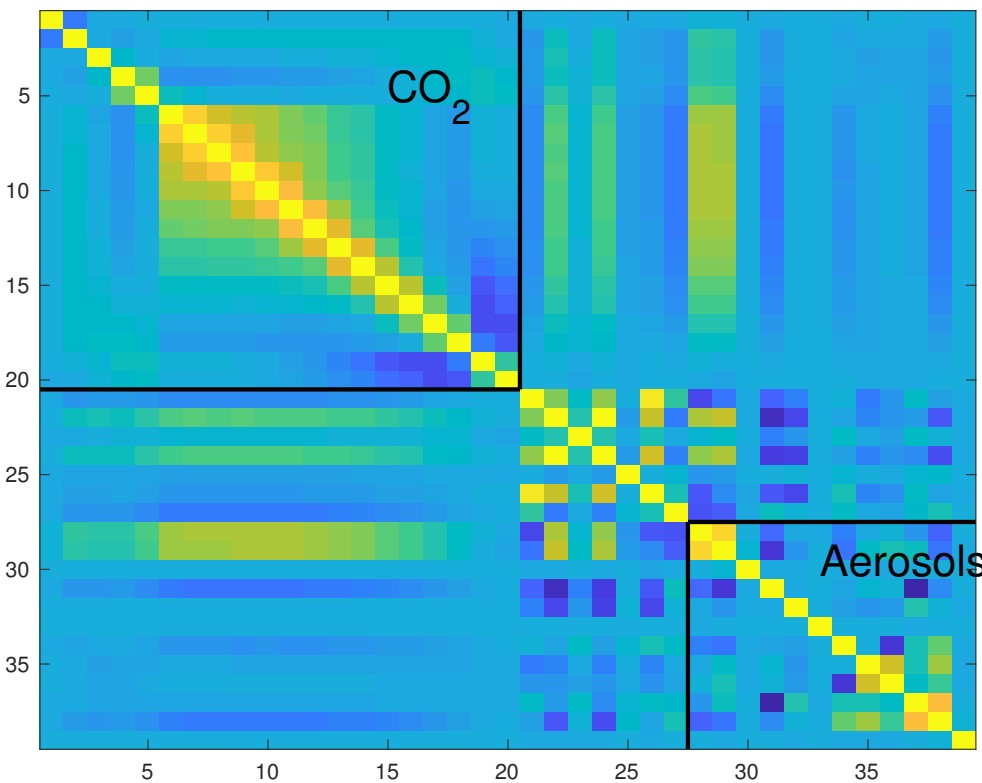

**Figure 7.** Posterior correlation matrix of $\hat{S}$. The elements of the matrix corresponding to $CO_2$ and aerosol parameters of the state vector are separated with black lines. It should be noted that the first aerosol parameters have a high correlation with the $CO_2$ part of the state vector.

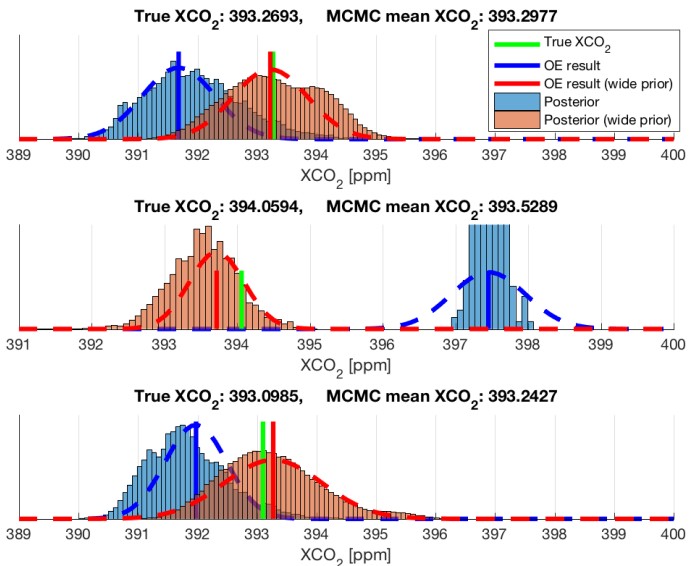

**Figure 8.** Three separate test cases showing $X_{CO_2}$ posterior histograms (using operational prior (blue) and widened prior) from LIS MCMC compared to the OE retrieval (blue and red dashed lines) and simulated ground truth (green). Also shown the true value of $X_{CO_2}$ with the corresponding MCMC mean using a widened prior. It should be noted that the example case 2 on the middle panel most likely did not converge in either the MCMC or OE when using the operational prior, which is not an unusual scenario even in the operational retrieval.

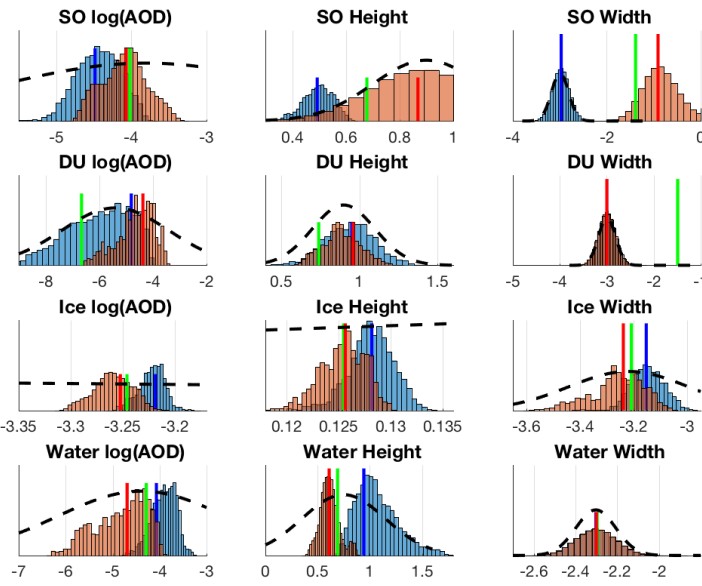

**Figure 9. Test case 1**: aerosol parameter marginal histograms (corresponding to retrievals using operational prior (blue) and widened prior (orange)) with operational prior (dashed black), ground truth (green) and OE retrieval result (blue and red vertical lines). SO = sulphate, DU = dust, Ice = cloud ice and Water = cloud liquid water.

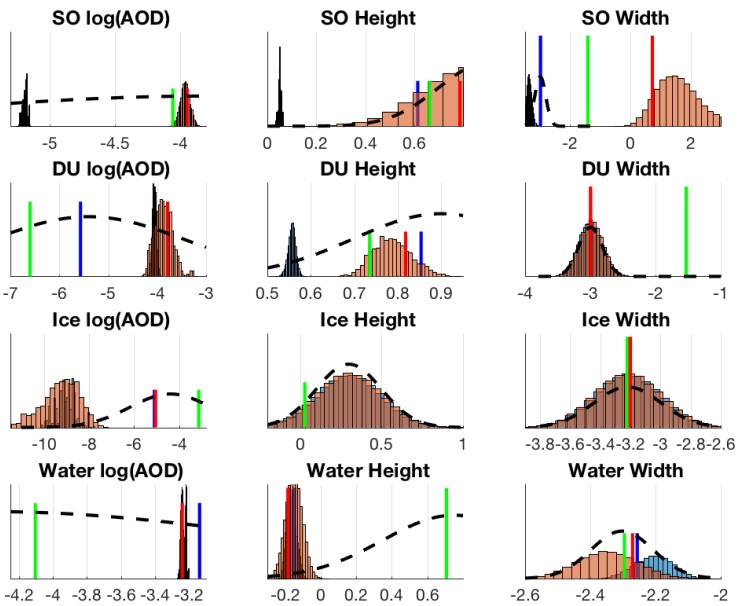

**Figure 10.** **Test case 2**: aerosol parameter marginal histograms (corresponding to retrievals using operational prior (blue) and widened prior (orange)) with operational prior (dashed black), ground truth (green) and OE retrieval result (blue and red vertical lines). SO = sulphate, DU = dust, Ice = cloud ice and Water = cloud liquid water.

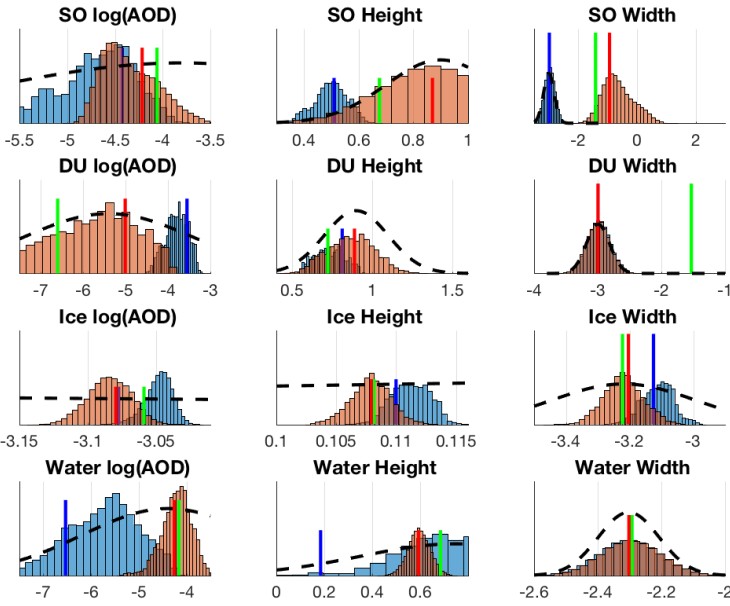

**Figure 11.** **Test case 3**: aerosol parameter marginal histograms (corresponding to retrievals using operational prior (blue) and widened prior (orange)) with operational prior (dashed black), ground truth (green) and OE retrieval result (blue and red vertical lines). SO = sulphate, DU = dust, Ice = cloud ice and Water = cloud liquid water.

## 4. Conclusions

Previous work on Monte Carlo experiments with the OCO-2 retrieval problem laid out the need to accelerate the convergence of these demanding computations. We have shown that in this case, where the problem is actually over-parametrized as the parameter space has a relatively low amount of informative directions, LIS dimension reduction can accelerate convergence of the MCMC simulation. In addition to the known fact that the $CO_2$ profile has only 2 degrees of freedom, the low information content of the aerosol part is also supported by recent research in e.g., [20]. The low information content is apparent when comparing the full-dimensional $X_{CO_2}$ MCMC chain (Figure 2) against the corresponding low dimensional LIS chain. Both were initiated at a value just outside the resulting posterior distribution, and upon visual inspection of the chains we can conclude that the LIS chain converges to the same result at least three times faster. Faster convergence is also supported by increased acceptance rate and effective sample size, as is to be expected thanks to the reduced dimension. This gain in computational efficiency makes it more feasible to conduct further studies with different geolocation templates.

The MCMC simulations allowed us to see the shape of the posterior distribution's aerosol and surface parameters that have repeatedly caused problems in previous work relating to the OCO-2 retrieval. The marginal distributions shown in Figure 4 show clear non-Gaussian shapes for some aerosol parameters, as well as the OE posterior being too narrow to contain the actual error limits. The most pathological cases are assembled in Figure 5 as 2D plots. In addition to showing again the inadequate coverage of $\hat{S}$, two observations can be made: first of all, parameters 31 and 37, both of which correspond to log(AOD), form a so-called "banana" distribution, which is a curved part of the distribution. This is known to cause problems for the adaptive MCMC (see e.g., [30]) which in part explains the low acceptance rates we obtained. Although we were mainly focused on aerosol parameters, we have included a plot of parameters 22 vs. 24 in this figure as well. These parameters correspond to albedo in the strong and weak $CO_2$ bands. As we can see from Figure 5, these parameters are strongly correlated, which is also a known feature that causes a "flat" posterior distribution which is harder to be characterized by MCMC algorithms.

The $X_{CO_2}$ histogram shown in Figure 2 suggests that, at least in this case, the well-tuned OE retrieval agrees quite well with MCMC in finding the posterior mean, but gives a too narrow uncertainty in comparison. This is in contrast to the findings of [16], but could be explained by OE not having converged to a global minimum. We have plotted the $X_{CO_2}$ distribution against the aerosol parameters in Figure 6, which further illustrates that OE underestimates the uncertainties propagated to $X_{CO_2}$ especially from the optical depth parameters of dust and cloud water. Although we have tested this with several other retrievals (see e.g., Figure 8), they were all from the same template with same dominating aerosol species and geophysical conditions. These results suggest there would be added value to conduct a comprehensive further thorough examination of different geolocation templates in the future.

Also, from panel 30 vs. 33 and from panels of SO and DU Coefficient 3 in Figure 4, we can see that the simulated ground truth is outside of the range of the operational prior distribution in the third parameter of both SO (sulphate) and DU (dust). This indicates that the retrieval tries to compensate the out-of-range-truth with other parameters and thus we might end up seeing possibly erroneous OE and MCMC results. As is apparent from Figure 7, these aerosol parameters (log AOD and profile height) are strongly correlated with the 20 $CO_2$ parameters and as such they can induce significant errors to the profile if retrieved incorrectly. Although this observation can in part explain the inconsistencies in the previous simulation experiments, it can also give a clue to the properties of the operational retrieval covariance matrix, since the state vectors used in the studies are based on real-world empirical physical parameters (see [17]). This also highlights the importance of prior validation and the use of MCMC to inspect the actual posterior distribution, since the OE approximation can give a misleading picture of the actual posterior uncertainties: a non-smooth likelihood function that might have a non-Gaussian

shape or multiple local minima, caused by e.g., the use of lookup tables in the model or the non-linear forward model.

Lastly, we perform the same MCMC retrieval for three separate test cases all having parameters 30 and 33 outside the range of the prior distribution, but with the difference that we inflate the prior covariance of above-mentioned SO Coefficient 3 parameter so that the resulting prior is practically non-informative in this direction and the state vector can vary freely. This naturally changes how the posterior distribution looks like, since the posterior is a product of the prior and likelihood distributions (see Equation (5)). The resulting $X_{CO_2}$ histograms are plotted in Figure 8 and we can clearly see that the ground truth is this time found roughly in the middle of the posterior distribution. Following [9,29], the parameters describing the prior width of distributions of aerosols 1 and 2 in the operational retrieval algorithm are intentionally left small. This way they are practically not retrieved, as the effect of the total aerosol profile is assumed to be compensated by the log(AOD) and aerosol height parameters. However, as this means that the resulting posterior does not contain the original ground truth in these selected test cases, we end up with a faulty posterior mean.

Allowing more variability with a wider prior covariance can result in finding the correct $CO_2$ concentration, but it still does not mean that the retrieved aerosol parameters correspond credibly to the ground truth. As can be seen from Figures 9–11 as some marginal histograms do actually converge to include the ground truth, some on the other hand seem worse after changing the prior. Our experiments also showed that this improved behavior is not generally observed in all retrievals, and factors such as OE convergence, simulated profile shape, other out-of-bounds parameters and even possible aerosol model problems (see e.g., [9] for added stratospheric aerosol) may lead to the OE and MCMC not agreeing with each other and/or the ground truth, as was observed in [16], even when the prior is relaxed.

## 5. Discussion

Results of this work open up several interesting topics for further research. Repeating the earlier experiments of [16] with LIS MCMC would probably aid in getting a better understanding of differences between different templates, as aerosols used in the state vector and their models change according to geolocation. In this work we have only seen the effect of dust and sulphate aerosol types, whereas there exist three more types in the operational OCO-2 retrieval algorithm: organic carbon, black carbon and sea salt. Investigating the combined effect of aerosols is also of interest, as different aerosol types seem to together affect the retrieval in a correlated and non-trivial way according to our results. To further assess the effect of aerosol parameters on $X_{CO_2}$ errors, a comprehensive study on several hundreds of different conditions should be conducted.

This work has demonstrated insights that can be gained with a strategic implementation of the LIS MCMC retrieval. The surrogate model illustrated here differs from the operational full physics retrieval in several ways, including the complexity of the state vector and radiance noise properties. A more efficient sampling algorithm also makes it more feasible to perform MCMC on the operational OCO-2 full physics model, and identify which features seen in this work are also present in the operational retrieval; although one might expect the results to be broadly applicable to the operational retrieval, there could be important differences not captured by the simplified state vector and forward model.

**Author Contributions:** Conceptualization, O.L., J.H., J.B., M.L., A.B. and J.T.; Data curation, O.L.; Formal analysis, O.L.; Funding acquisition, A.B. and J.T.; Investigation, O.L. and J.T.; Methodology, O.L., J.H., M.L. and J.T.; Project administration, A.B. and H.L.; Resources, J.H. and J.B.; Software, O.L., J.H. and J.B.; Supervision, M.L. and J.T.; Visualization, O.L.; Writing—original draft, O.L.; Writing—review & editing, O.L., J.H., J.B., M.L., J.T. and H.L.

**Funding:** This work has been financially supported by Finnish Academy Centre of Excellence in Inverse Modelling and Imaging (312125) and Academy of Finland project number 285421. Additional research was performed at the Jet Propulsion Laboratory, California Institute of Technology, under contract with NASA. Support was provided by the Orbiting Carbon Observatory-2 (OCO-2) mission.

**Acknowledgments:** The work presented here contributes to FMI activities on the OCO-2 Science Team. The authors thank Annmarie Eldering, Michael Gunson, and James McDuffie for valuable suggestions and technical assistance. We would also like to thank the two anonymous reviewers for their extremely useful comments and feedback.

**Conflicts of Interest:** 'The authors declare no conflict of interest. The funders had no role in the design of the study; in the collection, analyses, or interpretation of data; in the writing of the manuscript, or in the decision to publish the results.

## Abbreviations

The following abbreviations are used in this manuscript:

| | |
|---|---|
| AOD | Aerosol optical depth |
| ATBD | Algorithm Theoretical Basis Document |
| $CO_2$ | Carbon Dioxide |
| CS | Complement Subspace |
| LIS | Likelihood-Informed Subspace |
| MCMC | Markov chain Monte Carlo |
| MDPI | Multidisciplinary Digital Publishing Institute |
| OCO-2 | Orbiting Carbon Observatory 2 |
| OE | Optimal Estimation |
| $X_{CO_2}$ | Column averaged dry air mole fraction of $CO_2$ |

## Appendix A

*Appendix A.1. OE Convergence*

As was shown in [16,18], the optimal estimation retrieval seems to be sensitive to the first guess of the optimization algorithm, and furthermore might not agree with the overall posterior distribution obtained with MCMC. In order to rule out the possibilities of issues with the convergence of the Levenberg-Marquadt algorithm, we perform parameter tuning for the retrieval and compare the tuned version of the retrieval in our example case with the corresponding one used in [16,17].

In the previous work, the starting value for $\gamma$ parameter in the retrieval was set to 10 ([23]). Also, the tolerance for normalized step size was set to 4. To test the effect of these parameters, we set the maximum amount of iterations to 1000 so that the optimization will terminate when one of the other conditions is met. As a starting value for the optimization, we use the prior mean, the mean obtained from MCMC simulation, and the mean perturbed by $\pm\sigma \in \mathbb{R}^{39}$, where $\sigma$ is a standard deviation obtained by taking the square root of the elements of the diagonal of $\hat{S}$. The results are shown in the left panel of Figure A1 and Table A1. Next, we repeat the experiment by setting the starting value of $\gamma$ to 30 and reducing the tolerance of normalized step size to 0.0001. The results are shown in the right panel of Figure A1 for the $CO_2$ part and in and Table A2 for the aerosol part of the state vector.

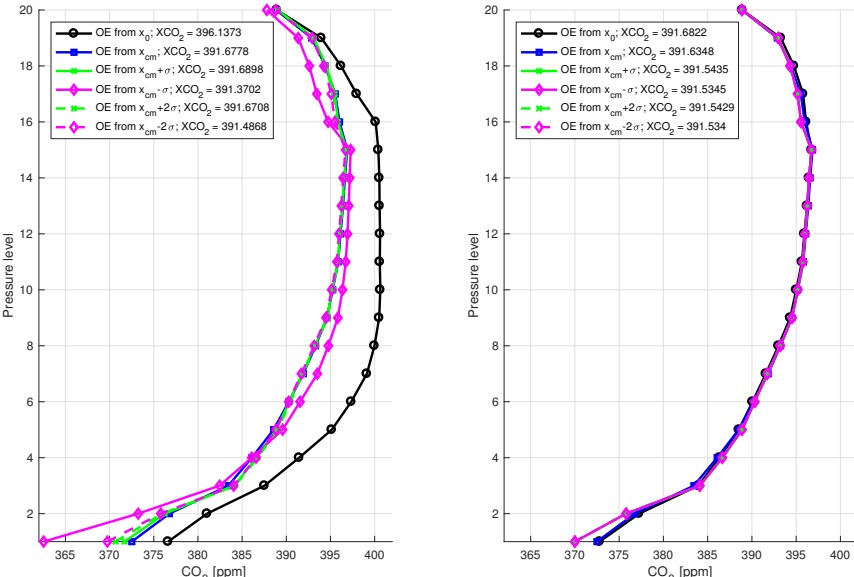

**Figure A1.** $CO_2$ profiles from OE retrievals with starting the optimization from different first guesses. **Left panel**: starting $\gamma = 10$, normalized step size tolerance = 40. **Right panel**: starting $\gamma = 30$, normalized step size tolerance = 0.0001.

**Table A1.** Aerosol parameter values and resulting $X_{CO_2}$ values from OE retrievals shown on the left panel of Figure A1. SO = sulphate, DU = dust, Ic = cloud ice and Wa = cloud liquid water.

|          | OE 1   | OE 2   | OE 3   | OE 4   | OE 5   | OE 6   |
|----------|--------|--------|--------|--------|--------|--------|
| SO 1     | −3.42  | −4.46  | −4.47  | −4.57  | −4.47  | −4.53  |
| SO 2     | 0.68   | 0.49   | 0.49   | 0.48   | 0.49   | 0.48   |
| SO 3     | −2.99  | −2.98  | −2.95  | −3.13  | −2.95  | −2.98  |
| DU 1     | −4.67  | e4.72  | −4.60  | −5.28  | −4.53  | −5.32  |
| DU 2     | 0.87   | 0.94   | 0.95   | 0.81   | 0.96   | 0.90   |
| DU 3     | −2.99  | −3.00  | −2.97  | −3.14  | −2.98  | −3.00  |
| WA 1     | −3.33  | −3.21  | −3.21  | −3.21  | −3.21  | −3.22  |
| WA 2     | 0.13   | 0.12   | 0.12   | 0.12   | 0.12   | 0.12   |
| WA 3     | −3.13  | −3.15  | −3.15  | −3.14  | −3.15  | −3.15  |
| IC 1     | −5.37  | −4.08  | −4.13  | −3.93  | −4.17  | −3.86  |
| IC 2     | 0.06   | 0.95   | 0.94   | 0.94   | 0.93   | 0.96   |
| IC 3     | −2.29  | −2.30  | −2.29  | −2.37  | −2.29  | −2.30  |
| $X_{CO_2}$ | 396.13 | 391.67 | 391.68 | 391.37 | 391.67 | 391.48 |

**Table A2.** Aerosol parameter values and resulting $X_{CO_2}$ values from OE retrievals shown on the right panel of Figure A1. SO = sulphate, DU = dust, Ic = cloud ice and Wa = cloud liquid water.

|          | OE 1   | OE 2   | OE 3   | OE 4   | OE 5   | OE 6   |
|----------|--------|--------|--------|--------|--------|--------|
| SO 1     | −4.48  | −4.48  | −4.52  | −4.52  | −4.52  | −4.52  |
| SO 2     | 0.49   | 0.49   | 0.48   | 0.48   | 0.48   | 0.48   |
| SO 3     | 2.97   | −2.98  | −2.97  | −2.97  | −2.97  | −2.97  |
| DU 1     | −4.80  | −4.72  | −4.96  | −5.08  | −4.97  | −5.12  |
| DU 2     | 0.95   | 0.94   | 0.94   | 0.93   | 0.94   | 0.93   |
| DU 3     | −2.99  | −3.00  | −2.99  | −2.99  | −2.99  | −2.99  |
| WA 1     | −3.21  | −3.21  | −3.22  | −3.22  | −3.22  | −3.22  |
| WA 2     | 0.12   | 0.12   | 0.12   | 0.12   | 0.12   | 0.12   |
| WA 3     | −3.15  | −3.15  | −3.15  | −3.15  | −3.15  | −3.15  |
| IC 1     | −4.06  | −4.06  | −3.95  | −3.92  | −3.95  | −3.91  |
| IC 2     | 0.94   | 0.95   | 0.95   | 0.95   | 0.95   | 0.95   |
| IC 3     | −2.30  | −2.30  | −2.30  | −2.30  | −2.30  | −2.30  |
| $X_{CO_2}$ | 391.68 | 391.63 | 391.54 | 391.53 | 391.54 | 391.53 |

*Appendix A.2. Surrogate Model State Vector*

**Table A3.** Surrogate forward model state vector description: element names and the prior values for the sounding performed in this study.

| No. | Name | Prior Value |
|---|---|---|
| | **CO2 Volume Mixing Ratio [ppm]** | |
| 1 | Vertical Level 1 (Top of Atmosphere) | 388.9123 |
| 2 | Vertical Level 2 | 393.7508 |
| 3 | Vertical Level 3 | 395.8462 |
| 4 | Vertical Level 4 | 397.4091 |
| 5 | Vertical Level 5 (Tropopause) | 398.5947 |
| 6 | Vertical Level 6 | 398.5761 |
| 7 | Vertical Level 7 | 398.5522 |
| 8 | Vertical Level 8 | 398.5237 |
| 9 | Vertical Level 9 | 398.4913 |
| 10 | Vertical Level 10 | 398.4556 |
| 11 | Vertical Level 11 | 398.4172 |
| 12 | Vertical Level 12 | 398.3766 |
| 13 | Vertical Level 13 | 398.3341 |
| 14 | Vertical Level 14 | 398.2900 |
| 15 | Vertical Level 15 | 398.2445 |
| 16 | Vertical Level 16 | 398.1977 |
| 17 | Vertical Level 17 | 398.1495 |
| 18 | Vertical Level 18 | 398.1000 |
| 19 | Vertical Level 19 | 397.7311 |
| 20 | Vertical Level 20 (Surface) | 397.2732 |
| 21 | Surface Pressure [hPa] | 1000 |
| | **Lambertian Albedo** | |
| 22 | Strong $CO_2$ Band Mean Albedo | 0.2296 |
| 23 | Strong $CO_2$ Band Albedo Spectral Slope | 0 |
| 24 | Weak $CO_2$ Band Mean Albedo | 0.2577 |
| 25 | Weak $CO_2$ Band Albedo Spectral Slope | 0 |
| 26 | $O_2$ A Band Mean Albedo | 0.2132 |
| 27 | $O_2$ A Band Albedo Spectral Slope | 0 |
| | **Aerosols** | |
| 28 | Sulphate Log Aerosol Optical Depth | −3.8054 |
| 29 | Sulphate Profile Height | 0.9000 |
| 30 | Sulphate Log Profile Thickness | −2.9957 |
| 31 | Dust Log Aerosol Optical Depth | −5.4027 |
| 32 | Dust Profile Height | 0.9000 |
| 33 | Dust Log Profile Thickness | −2.9957 |
| 34 | Cloud Ice Log Aerosol Optical Depth | −4.3820 |
| 35 | Cloud Ice Profile Height | 0.3000 |
| 36 | Cloud Ice Log Profile Thickness | −3.2189 |
| 37 | Cloud Water Log Aerosol Optical Depth | −4.3820 |
| 38 | Cloud Water Profile Height | 0.7500 |
| 39 | Cloud Water Log Profile Thickness | −2.3026 |

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
