# Peer review of "Accelerated MCMC for Satellite-Based Measurements of Atmospheric CO2"

_remotesensing, doi:10.3390/rs11172061_

Round 1
Reviewer 1 Report
This work moves forward the earlier study of Brynjarsdottir et al.(2018) by using the LIS MCMC framework to speed up the convergence of more traditional MCMC scheme. The authors arguably demonstrated that LIS is efficient in achieving the speedup of retrieving CO2 from OCO-2 measurements. I enjoyed reading this manuscript a lot, which makes me glad to see this work gets published after some minor revisions.
First, could the author explain why was linear transformations used while implementing the LIS dimension reduction? Any pitfalls with linearization?
Here are a few more specific comments to clarify the manuscript:
L67-68: can you justify this statement regarding aerosols’ impacts by providing references?
L85: a grammar mistake with ‘introduce’ after as well as. Should be ‘introducing’.
L88: change ‘chapter 4’ to ‘section 4’.
L146: a reference could be provided, though in the atmospheric science/remote sensing community such a solver is pretty common.
L168: change ‘transforms’ to ‘transformations’
Fig.5 & 6: it's better to at least list the variable names corresponding to the parameter numbers shown in these graphs to convey the info. to readers more directly.
In addition, if the authors could make the code performing the computation in this work open-access, that would be great, which may attract more readership.
Reviewer 2 Report
Review of “Accelerated MCMC for satellite-based measurements of atmospheric CO2”
By Lamminpää et al., 2019
They : 1) Speed up MCMC convergence significantly, enabling its use as a tool to
better understand the OCO-2 OE retrieval. 2) They look at the non-Gaussian shapes
of the posterior PDFs of XCO2. 3) They examine the effect of the operational prior
covariance of the aerosol parameters on the retrieval, in particular the vertical
width of an aerosol layer. 4) they can retrieve xco2 more accurately with this
analysis.
General Comments
Overall, I found this work interesting, though often the language was more
mathematical and less physically based. For instance, what is the physics involved
when the width of the sulfate layer is only loosely constrained; how does this allow
different values of the sulfate total AOD to be retrieved? But in general, the paper is
interesting, and shows explicitly that the posterior uncertainties on XCO2 given by
the operational OCO-2 retrieval algorithm are too low because of the nonlinearities
in the forward model – even the simplified forward model used here. The problem
could be even worse in the full operational retrieval, in which even more
parameters are retrieved.
One thing lacking in this work was that it was so dependent on just a few outlier
cases, the work gives little indication of how prevalent the problem of “wrong XCO2”
or “too low posterior uncertainty on XCO2” actually is with respect to the OE
retrieval. This can only be answered by performing their analysis on hundreds of
cases, rather than just a few. Or if they already did this, they should show a plot of
the results (in terms of XCO2 accuracy relative to truth or MCMC, as well as how
underestimated the XCO2 uncertainties are) across hundreds of geophysical cases.
That would greatly enhance the usefulness of this paper.
I recommend publishing this work after addressing the comments in this review.
Scientific Comments
Lines 24-26: TCCON has not historically been used for top-down flux estimation.
Please replace this with in-situ, flask, and aircraft measurements. Cite e.g. Peylin et
al (2013).
Lines 37-50: You’ll want to have some citations here. Rodgers (2000) for OE, O’Dell
et al (2012,2018) for the ACOS retrieval.
Line 70: Please define “acceptance ratio” for non-MCMC experts.
Line 31-2: [4] (Miller et al, 2007) says no such thing. Only section 3.2 of that work
discusses the effect of biases, and finds biases of a few tenths of a ppm can have
significant deleterious impacts on the inferred CO2 fluxes. It is generally believed in
the community that biases must be below a few tenths of a ppm regionally and
seasonally to enable accurate deduction of carbon fluxes.
Section 2.2: You should state that this approach is missing several parameters in the
operational state vector : 9 EOF amplitudes, 6 dispersion parameters, a temperature
offset parameter, a water vapor parameter, and solar-induced fluorescence (SIF) in
the O2A-band. You may expect the results to be broadly applicable to the
operational retrieval, but there could be important differences not captured by the
simplified state vector and forward model. To know for sure, you’d need to a
limited MCMC experiment on the operational forward model.
Section 2.3, regarding the noise model: There is an important difference between
your proposed “constant SNR-within-a-band” noise model and how OCO-2 actually
works. OCO-2 has smaller noise in line cores, and larger noise in the continuum.
However, the SNR is smaller in line cores and larger in the continuum because of the
quadratic nature of the noise model. This may also limit the applicability of the
results; you do not show that you get the same qualitative results with this
simplification. It’s confusing why you made this choice, because using a more
accurate noise is straightforward and likely would have not slowed down your
calculations.
Section 2.3. If you want this work to be read and understood by retrieval people, it
would be better to use their notation. That is:
• S (instead of Σ) for covariance matrices
• K (instead of J) for the Jacobian of the forward model
• P (instead of π) for a PDF
“J” in particular is confusing, as this is nearly always used for “cost function” in
the atmospheric community.
Line 250: Instead of “is not designed for profile retrieval”, you might say that
although the OCO-2 algorithm retrievals a full CO2 profile on 20 levels, it only
validates its full-column retrievals, as there are typically only two DOFs for CO2
information in the retrieved profile.
Section 3.2: The prior constraint on the aerosol width is so tight that this parameter
is effectively not retrieved. This was based on early work (Butz et al, Applied Optics,
Vol 48, Issue 18, 2009) which showed that the spectra had little sensitivity to it.
Please make very clear that the prior uncertainty is very tight and that this
parameter is effectively not retrieved. It is fascinating that opening up the prior
uncertainty (section 3.3) has a large effect on some cases. You may wish to point
out that this is a test that could be explored in the operational retrieval.
Further, because so much of this paper is about aerosols, you may wish to point out
that in general, the retrieval has between 2-6 DOFs for aerosol quantities (e.g.
Nelson et al., 2019) out of the 12 total aerosol parameters you include.
Figure 4: The labels are meaningless to most people. Suggest you replace
“Coefficient 1” with Ln(AOD) and Coefficient 2 as “Height (P/Psurf)” or something
more descriptive & meaningful.
Figure 5: Are the two blue contours the 68% and 95% CLs? Please add this
information to the figure caption.
Figure 6: can you also include the prior curves? I’m confused for instance by how
tight the range is on parameter 34, Ln(Ice AOD), but the truth is still outside the
MCMC distribution.
Figure 10 (test case 2): This has the largest XCO2 discrepancy between retrieved
and truth. The widened prior uncertainty yielded a SO Coefficient 3 distribution
that is between 0 and 2, with a mean around 1 (and the widened OE around 0.7).
This puts the actual width of the distribution at around 2, which is in units of
P/Psurf. This physically makes little sense. How does the retrieval forward model
actually deal with this? Does it only include the portion of the aerosol distribution
between 0 and 1? Further, the operational retrieval only retrieves the width of the
aerosol layer, not the natural log of the width. Why did you make this change from
the operational retrieval? Would you results change much if you matched to what
the operational retrieval does?
Technical Comments
Line 26: “relating to the carbon cycle and in particular CO2 emissions”
Line 202: with standard à with the standard
Line 242: using same sample à using the same sample
Line 248: This choice lead to a better à This choice led to a better
Line 254: works really well à works well; succeeds to capture à succeeds in
capturing
Line 259: add a comma after “full posterior distribution”
Line 260: “peak point” à “mode of the posterior distribution” or “peak of posterior
distribution”; “expectation” à “expected value” (though googling it, I see
“expectation” is also acceptable; I usually hear the latter)
Line 280: Fix duplicate “5”
References:
Nelson, R. R. and O'Dell, C. W.: The impact of improved aerosol priors on near-infrared
measurements of carbon dioxide, Atmos. Meas. Tech., 12, 1495-1512,
https://doi.org/10.5194/amt-12-1495-2019, 2019.
Peylin, P., et al., “Global atmospheric carbon budget: results from an ensemble of
atmospheric CO2 inversions,” Biogeosciences, 10, 6699–6720, https://doi.org/10.5194/bg-
10-6699-2013, 2013.
